# Evaluation of HPV-Related Biomarkers in Anal Cytological Samples from HIV-Uninfected and HIV-Infected MSM

**DOI:** 10.3390/pathogens10070888

**Published:** 2021-07-13

**Authors:** Francesca Rollo, Alessandra Latini, Massimo Giuliani, Amalia Giglio, Maria Gabriella Donà, Maria Benevolo

**Affiliations:** 1Pathology Department, Regina Elena National Cancer Institute IRCCS, 00144 Rome, Italy; francesca.rollo@ifo.gov.it (F.R.); maria.benevolo@ifo.gov.it (M.B.); 2STI/HIV Unit, San Gallicano Dermatological Institute IRCCS, 00144 Rome, Italy; massimo.giuliani@ifo.gov.it (M.G.); mariagabriella.dona@ifo.gov.it (M.G.D.); 3Microbiology and Clinical Pathology Department, San Gallicano Dermatological Institute IRCCS, 00144 Rome, Italy; amalia.giglio@ifo.gov.it

**Keywords:** human papillomavirus, MSM, HIV, anal cytology, DNA, mRNA, p16/Ki-67, dual staining, anal cancer, screening

## Abstract

Men who have sex with men (MSM) harbor the highest risk for anal carcinoma, mainly caused by Human Papillomavirus (HPV). The use of HPV-related biomarkers in the screening for this neoplasia is still debated. We assessed the association between high-risk (hr)HPV DNA, HPV16/18 DNA, hrHPV E6/E7 mRNA, and p16/Ki-67 with cytological abnormalities (any grade) and high-grade intraepithelial lesions (HSIL) in HIV-uninfected and HIV-infected MSM. Overall, 150 cytological samples in PreservCyt (Hologic), with a negative to HSIL report, were analyzed for hrHPV DNA, hrHPV E6/E7 mRNA, and p16/Ki-67 using the Linear Array (Roche), Aptima (Hologic), and CINtec^®^ PLUS (Roche) assays. In HIV-infected MSM, positivity for all the biomarkers significantly increased with the cytological grade. In both populations, the association of hrHPV E6/E7 mRNA and p16/Ki-67 positivity with HPV16 did not differ significantly compared to hrHPVs other than HPV16. In HIV-uninfected MSM, the odds of having an HSIL increased approximately six times for the p16/Ki-67 positive cases. In HIV-infected individuals, all the biomarkers showed a significant association with HSIL, except for hrHPV DNA, with the strongest association observed for p16/Ki-67. The odds of HSIL increased almost 21 times in those positive for this biomarker. Our results encourage further investigation on the use of p16/Ki-67 dual staining in anal cancer screening for HIV-uninfected and HIV-infected MSM.

## 1. Introduction

Anal cancer represents a rare neoplasm, with approximately 50,000 new cases estimated in 2020 worldwide [1]. Despite being uncommon, the number of cases is estimated to increase up to 78,000 in 2040 [1]. Men who have sex with men (MSM) show a higher risk than the general population for this neoplasia, with the highest incidence observed among MSM with HIV infection [2]. Human papillomavirus (HPV) is recognized as a causal agent of anal squamous cell carcinomas [3]. It is responsible for a large majority of the cases, with the predominant genotype being HPV16 in both HIV-positive and HIV-negative men [4,5]. Prophylactic HPV vaccination of pre-teens represents a powerful tool for the prevention of future anal cancers, but other preventive measures are needed for adult high-risk subjects. Anal cancer screening has been adopted for at-risk populations and guidelines from regional or national scientific societies have been specifically issued for HIV-infected subjects and MSM. Analogies between morphology features of the anal canal and the cervix, as well as the risk factors for cancer development in these anatomical regions, have led to the use of the strategies employed for cervical cancer screening also for anal cancer screening. Although the natural history of anal cancer is not as clear as that of cervical cancer, it seems that high-grade squamous intraepithelial lesions (HSIL based on the Lower Anogenital Squamous Terminology-LAST) precede the development of cancer [6]. Despite the fact that the randomized trial to evaluate whether detection and treatment of HSIL can lead to anal cancer prevention in HIV-infected MSM is still ongoing [7], these individuals are subjected to anal cancer screening in some clinical settings. However, the optimal screening modality has not been identified yet, so that the algorithms used are local and based on individual expert opinion [8,9,10]. The International Anal Neoplasia Society recommended an algorithm of screening based on digital ano-rectal examination (DARE), anal cytology and high-resolution anoscopy (HRA) [11]. Histological evaluation of HRA-guided biopsies is considered as the gold standard for the diagnosis of anal pre-cancerous and cancerous lesions. However, at present, HRA is not widely available, even in clinical settings attended by at-risk individuals [12,13,14]. Anal cytology represents the most common screening option, being technically easy, low-cost, and highly accessible. A recent meta-analysis reported for anal cytology showed a pooled sensitivity of 85.0% and a pooled specificity of 43.2% for the detection of HSIL or worse lesions [15]. However, cytology is subjective and has low reproducibility. Moreover, the cytological categories showed different predictive values in the identification of histological HSILs [16]. In order to improve the detection of these lesions and to use more objective methods, several HPV-related biomarkers have been evaluated, mainly in HIV-infected MSM [17,18,19,20,21,22]. High-risk (hr)HPV E6/E7 mRNA (thereafter hrHPV mRNA) has gained attention because it indicates an active and transforming HPV infection. Another biomarker under study is the cyclin-dependent kinase inhibitor 2A, i.e., p16INK4a (p16), since its over-expression is induced by E7 viral oncoprotein. Currently, an immunostaining assay is available for the simultaneous assessment of p16 expression together with Ki-67, a well-established proliferation marker. Since the co-expression of these proteins does not occur under physiological conditions, because they exert opposite effects on the cell cycle, their concomitant expression within the same cell may serve as an indicator of the HPV-induced deregulation of the cell cycle. However, the use of these HPV-related biomarkers still needs to be validated in anal cancer screening, thus further investigations are needed, especially in HIV-uninfected MSM. As far as we know, there are only a few studies that compared simultaneously several biomarkers in both HIV-infected and uninfected MSM [23,24]. In the present study, we evaluated anal cytological samples collected from HIV-infected and uninfected MSM to assess the distribution of hrHPV DNA, HPV16 and/or 18 (HPV16/18) DNA, hrHPV mRNA, and p16/Ki-67 according to the cytological report, and their association with cytological abnormalities.

## 2. Results

### 2.1. Study Population

Overall, 150 anal samples were included in the study. They had been obtained between August 2014 and December 2019 from 47 HIV-uninfected MSM (median age: 38 years, interquartile range [IQR]: 29–48) and 103 HIV-infected (median age: 40.5 years, IQR: 35–48). Of the HIV-infected subjects, 97.1% were on combined antiretroviral therapy (cART). The distribution of the cytology reports for the two study groups was as follows: (i) HIV-uninfected MSM: 14 (29.8%) negative for intraepithelial lesion or malignancy (NILM), 9 (19.1%) atypical squamous cells of undetermined significance (ASCUS), 11 (23.4%) low-grade squamous intraepithelial lesion (LSIL), 13 (27.7%) HSIL; (ii) HIV-infected MSM: 27 (26.2%) NILM, 26 (22.4%) ASCUS, 31 (30.1%) LSIL, 23 (22.3%) HSIL.

### 2.2. Distribution of the Biomarker Positivity by Cytological Report

All the samples analyzed gave a valid result for hrHPV mRNA, whereas two samples were inadequate for p16/Ki-67 interpretation (one ASCUS and one LSIL in the HIV-uninfected subgroup). The distribution of the positivity for the four biomarkers under study according to the cytological report and the HIV status is shown in Table 1.

In HIV-uninfected MSM, none of the investigated biomarkers showed a significantly increasing trend of positivity rate with the degree of the cytological report. Nonetheless, the positivity rate of hrHPV DNA, HPV16/18 DNA, and p16/Ki-67 was the highest in HSILs (84.6%, 46.2%, and 84.6%, respectively). In HIV-infected MSM, we observed a significant increase in the positivity rate from NILM to HSIL for each biomarker, with the same positivity in HSIL for all the biomarkers (95.7%) except for HPV16 and/or HPV18 DNA (73.9%).

### 2.3. Association of hrHPV mRNA and p16/Ki-67 with HPV16

To evaluate whether the positivity for hrHPV mRNA and p16/Ki-67 was influenced by hrHPV genotypes, we assessed their association with the presence of HPV16, the most carcinogenic and prevalent type in anal cancer, versus the hrHPVs other than HPV16, separately for HIV-uninfected and infected MSM (Table 2).

In both study groups, more than 90.0% of the HPV16-positive samples were mRNA-positive, and nearly 77.0% showed p16/Ki-67 staining. Positivity for hrHPV mRNA or p16/Ki-67 did not display a significantly different association with the presence of HPV16 DNA compared to the presence of hrHPVs other than HPV16 DNA. 

### 2.4. Association of the Anal Cytological Abnormalities with the Biomarkers

Table 3 shows the association of the cytological abnormalities at the ASCUS+ and HSIL threshold with the four single biomarkers, for HIV-uninfected and HIV-infected MSM, separately. None of the biomarkers resulted in being significantly associated with ASCUS+ in HIV-uninfected MSM (Table 3A). Differently, we observed a significant association of ASCUS+ and hrHPV DNA (*p* = 0.004), HPV16/18 DNA (*p* = 0.025), hrHPV mRNA (*p* = 0.0001), and p16/Ki-67 positivity (*p* < 0.0001) in HIV-infected MSM (Table 3B). In particular, the strongest association was found for p16/Ki-67 (Odds Ratio [OR]: 10.50, 95% Confidence Interval [CI]: 3.69–29.90). This biomarker was also the only one that showed a significant association with HSIL in HIV-uninfected MSM (*p* = 0.031). Instead, within the HIV-infected subgroup, the odds of an HSIL report increased significantly in those positive for HPV16/18 DNA (*p* = 0.008), hrHPV mRNA (*p* = 0.02), and p16/Ki-67 (*p* = 0.004), with the strongest association observed for the latter biomarker (OR: 20.93, 95% CI: 2.69–162.79).

We then assessed the association of the cytological abnormalities at the ASCUS+ and HSIL threshold with combinations in pairs of the investigated biomarkers, evaluating the cases positive for any biomarker of the couple and, separately, those showing simultaneous positivity for both biomarkers of the couple. Table 4 shows the results separately for the two study groups. In HIV-uninfected MSM, ASCUS+ and HSIL reports did not show significant associations with any of the biomarker combinations (Table 4A). In HIV-infected subjects (Table 4B), the combinations that included p16/Ki-67 displayed the strongest associations with ASCUS+ in the case of double positivity (OR in the range of 17.77 and 20.77). Regarding HSIL, the strongest associations were observed in the case of simultaneous positivity for p16/Ki-67 in combination with HPV16/18 DNA (OR: 41.80) or hrHPV mRNA (OR: 25.80).

## 3. Discussion

In this study, we evaluated four biomarkers, i.e., hrHPV DNA, HPV16/18 DNA, hrHPV E6/E7 mRNA, and p16/Ki-67 in anal cytological samples from HIV-uninfected and HIV-infected MSM. 

For all the biomarkers, we observed that, in HIV-infected MSM, the positivity rate increased significantly with the grade of the cytological alterations, consistently with previous studies [21,25,26] and a recent meta-analysis [4]. Differently, we did not observe this trend in HIV-uninfected MSM, probably due to the limited number of samples evaluated for this subgroup. Overall positivity for hrHPV mRNA was around 70.0%, quite close to that for hrHPV DNA, in both study groups. Our mRNA positivity rate was higher compared to other studies conducted on MSM [20,23,25]. However, they used the Pretect HPV Proofer/NucliSENS assay, which only detects E6/E7 mRNA of HPV 16, 18, 31, 33, and 45. Given that the Aptima assay allows the detection of the mRNA of 14 hrHPVs, this may be the reason why mRNA positivity was higher in our patients. Our mRNA positivity was also higher compared to a study that used the Aptima test (70% versus 50.0%), but their study population was not exclusively composed by MSM [17].

Overall positivity for p16/Ki-67 staining in our study was about 60.0%, very close to that reported by others [20,23,25]. Interestingly, in our study groups, a considerable proportion of ASCUS and LSIL cases were positive for p16/Ki-67. We cannot exclude that these cases corresponded to HSIL on histology. Indeed, it has been shown that HIV-infected and uninfected MSM with an ASCUS or LSIL report on cytology may have an HSIL or worse lesion on histology [27]. Therefore, p16/Ki-67 may suggest which ASCUS or LSIL cases likely correspond to histological HSILs. 

Notably, three HSILs were negative for both hrHPV DNA and mRNA. One case was positive for the possibly carcinogenic type HPV53, and, interestingly, also for p16/Ki-67 staining. Similarly, cervical HSILs have been found that are caused by possibly carcinogenic types, and, consequently, they test negative for both hrHPV DNA and mRNA [28]. The remaining two cases were negative for all the HPV types detectable by the Linear Array (one of them displayed the dual staining). This may suggest either the presence of uncommon HPVs, not revealed through the tests employed in the study, or the lack of the target region within the L1 gene, which produced a false negative result with the HPV DNA test. Unfortunately, we could not confirm these HSILs given the lack of corresponding histology.

We observed that the rate of hrHPV mRNA and p16/Ki-67 positivity was similar in the cases harboring HPV16 or hrHPV genotypes other than HPV16. The strength of the association of HPV16 with hrHPV mRNA and p16/Ki-67 positivity did not appear to be significantly different compared to hrHPV types other than HPV16. Therefore, hrHPV mRNA and p16/Ki-67 may be useful also when the anal infection is caused by hrHPV types other than HPV16.

We then evaluated the individual biomarkers in terms of association with ASCUS+ and HSIL cytological reports. Our findings showed that all of them were significantly associated with ASCUS+ in HIV-infected MSM, whereas a lack of significant associations was observed for the HIV-uninfected counterparts. Most importantly, the odds of an HSIL report increased six times and almost 21 times in HIV-uninfected and HIV-infected MSM who were positive for p16/Ki-67, respectively. In a recent meta-analysis, pooled sensitivity and specificity of 56.6% and 62.3%, respectively, have been reported for this biomarker in MSM [29]. However, a large variation has been observed for the performance of p16/Ki-67. Indeed, a sensitivity equal to or above 90.0% has been reported in a few studies [20,23,25].

Among HIV-infected MSM, the odds of having an ASCUS+ and HSIL report were approximately seven and eleven times higher in hrHPV mRNA-positive subjects. Testing for mRNA is generally more specific than DNA [20,25,26,30,31] with a similar [26,30] or lower sensitivity [20], as also confirmed by a recent meta-analysis [29].

HIV-infected MSM positive for hrHPV DNA showed significantly increased odds of having an ASCUS+ report, whereas the association did not reach statistical significance for the HSIL report. A very recent study showed that hrHPV DNA testing displays a very high sensitivity (96%) but very low specificity (23–27%) for HSIL or worse lesions in both HIV-infected and uninfected MSM [9], in line with previous findings [29]. The very high background prevalence of hrHPV DNA in the anal canal of MSM and the fact that most of the anal infections are transient and not associated with pre-cancer or cancer will likely limit the utility of hrHPV DNA testing in this at-risk population. A significant association of an ASCUS+ or HSIL report with HPV16/18 DNA positivity was observed among HIV-infected MSM, despite the fact that the odds were lower compared to hrHPV DNA. HPV typing is more specific than hrHPV testing, despite being less sensitive, and does not provide enough immediate or long-term reassurance against the risk of HSIL or worse lesions [20].

Regarding the combinations of the biomarkers, none of the investigated combinations showed a significant association with ASCUS+ or HSIL among HIV-uninfected MSM. Conversely, the odds of having ASCUS+ cytology for HIV-infected MSM significantly increased when p16/Ki-67 was in combination with any of the other biomarkers, and the two biomarkers of the pair were simultaneously positive. The strength of these associations was higher than in the case of p16/Ki-67 used as a single biomarker. In addition, the double positivity for p16/Ki-67 and HPV16/18 DNA increased the odds of an HSIL report by almost 2-fold in comparison with its use as a single biomarker. These results suggest that, while p16/Ki-67 alone could be useful to identify the subjects with the highest odds of having a high-grade cytological lesion among HIV-uninfected MSM, its combination with other biomarkers, in particular with HPV 16/18 DNA, improves the possibility to identify those more likely to have an HSIL among HIV-infected MSM.

The differences observed between the two study groups are not surprising. The performance of HPV-related biomarkers generally differs by HIV status [9]. Anal cytology itself has a higher sensitivity in HIV-infected compared to HIV-uninfected MSM, since HIV-infected individuals usually have larger and multiple lesions, that are more likely detectable by cytology [32].

The main limitation of our study is represented by the lack of histological diagnosis to confirm the presence and grade of the lesions. The analysis was limited to the cytological abnormalities, which may not reflect the histological diagnosis, especially in terms of lesion grade. Differently, the strengths of this study are represented by the concomitant evaluation of four different biomarkers and the inclusion of both HIV-infected and uninfected MSM, although the number of cytological samples from the latter individuals could have affected the power of some estimates in this subgroup.

In conclusion, we evidenced that among the single biomarkers, p16/Ki-67 displayed the strongest association with cytological abnormalities of any grade in HIV-infected MSM and with HSIL cytology in both groups. The combinations did not show to be useful with respect to p16/Ki-67 alone among HIV-uninfected MSM, whereas they substantially improved the strength of association with ASCUS+ and HSIL when p16/Ki-67 was used together with HPV 16/18 DNA among HIV-infected MSM. Further studies are necessary to better understand the usefulness of these biomarkers in anal cancer screening, either for primary testing or triage of abnormal cytology, especially considering the limited availability of HRA and HRA-trained clinicians.

## 4. Materials and Methods

### 4.1. Study Population and Sample Collection

Study participants were represented by MSM aged ≥18 years, attending the center for Sexually Transmitted Infections (STIs) and HIV of the San Gallicano Dermatological Institute (Rome, Italy) for the Surveillance Program of Anal Intraepithelial Neoplasia (SAIN project) [33]. This project aimed to assess the prevalence, incidence, and determinants of anal HPV infection and anal lesions in HIV-infected and uninfected MSM. The participants were subjected to anal cytological sampling by means of a sterile Dacron swab as previously detailed [34]. Briefly, the collected cells were suspended in PreservCyt (Hologic), and starting from 2014, the residual sample remaining after HPV DNA testing and the anal Pap test was stored at +4 °C. For the purposes of the present study, these specimens were evaluated based on the following criteria: 1. Sample adequate for morphological interpretation; 2. results available for HPV DNA testing; 3. residual sample sufficient to perform the assays of interest for this study. Among the 480 samples with the inclusion criteria, 150 were selected in order to homogeneously represent all the cytological categories. No cases with a cancer diagnosis were available. 

### 4.2. hrHPV DNA and hrHPV E6/E7 mRNA Testing

The presence of HPV DNA was detected by the Linear Array HPV Genotyping Test (Roche Diagnostics, Milan, Italy) according to the manufacturer’s instructions [34]. This PCR-based assay can also detect the HPV genotypes classified as high-risk by IARC [3]. Based on the results of the HPV DNA test, samples were classified as positive or negative for: (i) hrHPV DNA (HPVs 16, 18, 31, 33, 35, 39, 45, 51, 52, 56, 58, 59, 66, 68); (ii) HPV16/18 DNA. The evaluation of hrHPV E6/E7 mRNA was achieved by the Aptima HPV Assay (Hologic, Pomezia, Italy), which allows the detection of the E6/E7 mRNA of 14 high-risk genotypes (16, 18, 31, 33, 35, 39, 45, 51, 52, 56, 58, 59, 66, 68) as a pooled result. One milliliter of the specimens was transferred into an Aptima Specimen Transfer tube (Hologic). Isolation of the mRNA target and detection of the RNA amplicons were performed automatically by the Panther System (Hologic). An Internal Control is present to monitor the adequacy of each step. 

### 4.3. Liquid-Based Cytology

The slides were obtained by ThinPrep 2000 processor (Hologic) and stained using the Papanicolaou protocol. The morphology had been classified by a certified cytopathologist following the Bethesda guidelines [35,36].

### 4.4. Immunocytochemistry for p16/Ki-67

Slides for immunocytochemical staining were prepared using a ThinPrep 2000 processor (Hologic). The expression of p16 (clone E6H4™) and Ki-67 (clone 274-11 AC3) was evaluated using CINtec^®^ PLUS Cytology Kit (Roche Diagnostics) following the manufacturer’s instructions. The samples were considered positive when at least one cell showed simultaneous staining for p16 and Ki-67. The immunostaining was considered as inadequate for interpretation in the case of p16 background and/or weak counterstaining.

### 4.5. Data Analysis

In order to summarize all the variables of interest of the study population, descriptive statistics were used. The distribution of the positivity for each biomarker according to the cytological category was assessed separately for HIV-uninfected and infected individuals. To assess the association of the positivity for hrHPV mRNA and for p16/Ki-67 with HPV16, the HPV DNA results were classified as positive for HPV16 or “hrHPVs other than HPV16” (18, 31, 33, 35, 39, 45, 51, 52, 56, 58, 59, 66, 68). ORs together with their 95% CI and p value from z statistics were calculated. ORs were also used to evaluate the association of the investigated biomarkers, individually and in selected combinations, with anal cytological abnormalities using as a cut-off point any abnormality (i.e., ASCUS+ threshold) and HSIL. The investigated biomarkers were combined in pairs and the cases were classified as: Negative for both biomarkers, positive for any of the two biomarkers, positive for both biomarkers. A *p* value <0.05 was considered as significant. Statistical analyses were carried out using MedCalc Statistical Software version 19.3.1.

## Figures and Tables

**Table 1 pathogens-10-00888-t001:** Positivity rate of the investigated biomarkers according to the cytological report and stratified by HIV status.

	Positivity n (%)					
**HIV-Uninfected MSM**	**NILM** **n = 14**	**ASCUS** **n = 9**	**LSIL** **n = 11**	**HSIL** **n = 13**	**Total** **n = 47**	***p* for Trend ^1^**
hrHPV DNA	10 (71.4)	6 (66.7)	8 (72.7)	11 (84.6)	35 (74.5)	0.78
HPV16/18 DNA	6 (42.9)	4 (44.4)	4 (36.4)	6 (46.2)	20 (42.6)	0.97
hrHPV mRNA	8 (57.1)	5 (55.6)	9 (81.8)	10 (76.9)	32 (68.1)	0.42
p16/Ki-67 ^2^	6 (42.9)	3 (37.5)	6 (60.0)	11 (84.6)	26 (57.8)	0.09
**HIV-Infected MSM**	**NILM** **n = 27**	**ASCUS** **n = 22**	**LSIL** **n = 31**	**HSIL** **n = 23**	**Total** **n = 103**	***p* for Trend ^1^**
hrHPV DNA	16 (59.3)	17 (77.3)	27 (87.0)	22 (95.7)	82 (79.6)	0.009
HPV16/18 DNA	8 (29.6)	7 (31.8)	18 (58.1)	17 (73.9)	50 (48.5)	0.004
hrHPV mRNA	12 (44.4)	14 (63.6)	28 (90.3)	22 (95.7)	76 (73.8)	<0.0001
p16/Ki-67	6 (22.2)	14 (63.6)	21 (67.7)	22 (95.7)	63 (61.2)	<0.0001

^1^ chi-square test for trend from NILM to HSIL. ^2^ percentages were calculated over the number of valid samples for this biomarker, i.e., 14 NILM, 8 ASCUS, 10 LSIL, 13 HSIL, for a total of 45 cases. NILM, negative for intraepithelial lesion or malignancy; ASCUS, atypical squamous cells of undetermined significance; LSIL, low-grade squamous intraepithelial lesion; HSIL, high-grade squamous intraepithelial lesion; hr, high-risk; MSM, men who have sex with men.

**Table 2 pathogens-10-00888-t002:** Association of hrHPV mRNA and p16/Ki-67 positivity with HPV16 DNA in comparison with “hrHPVs other than HPV16” DNA.

	HIV-Uninfected MSM			
HPV DNA	hrHPV mRNA n/N (%)	OR (95% CI); *p* Value	p16/Ki-67 n/N (%)	OR (95% CI); *p* Value
hrHPVs other than HPV16	17/21 (80.9)	Ref	13/21 (61.9)	Ref
HPV16	13/14 (92.9)	3.06 (0.30–30.73); 0.34	10/13 (76.9)	2.05 (0.43–9.78); 0.37
	**HIV-Infected MSM**			
hrHPVs other than HPV16	34/39 (87.2)	Ref	25/39 (64.1)	Ref
HPV16	41/43 (95.3)	3.01 (0.55–16.53); 0.20	33/43 (76.7)	1.85 (0.70–4.84); 0.21

hrHPVs other than HPV16: 18, 31, 33, 35, 39, 45, 51, 52, 56, 58, 59, 66, 68; CI, Confidence Interval; OR, Odds Ratio; Ref, reference.

**Table 3 pathogens-10-00888-t003:** Association of ASCUS+ and HSIL cytology with the single biomarkers in (A) HIV-uninfected and (B) HIV-infected MSM.

**(A)**	**HIV-Uninfected MSM**					
	**ASCUS+**			**HSIL**		
	**n/N** **(%)**	**OR** **(95% CI)**	***p* Value**	**n/N** **(%)**	**OR** **(95% CI)**	***p* Value**
**hrHPV DNA**						
**−**	8/12 (66.7)	Ref		2/12 (16.7)	Ref	
**+**	25/35 (71.4)	1.25 (0.31–5.10)	0.76	11/35 (31.4)	2.29 (0.43–12.27)	0.33
**HPV16/18 DNA**						
**−**	19/27 (70.4)	Ref		7/27 (25.9)	Ref	
**+**	14/20 (70.0)	0.98 (0.28–3.48)	0.98	6/20 (30.0)	1.22 (0.34–4.43)	0.76
**hrHPV mRNA**						
**−**	9/15 (60.0)	Ref		3/15 (20.0)	Ref	
**+**	24/32 (75.0)	2.54 (0.54–7.39)	0.30	10/32 (31.2)	1.82 (0.42–7.90)	0.42
**p16/Ki-67**						
**−**	11/19 (57.9)	Ref		2/19 (10.5)	Ref	
**+**	20/26 (76.9)	2.42 (0.67–8.80)	0.18	11/26 (42.3)	6.23 (1.19–32.75)	0.031
**(B)**	**HIV-Infected MSM**					
	**ASCUS+**			**HSIL**		
	**n/N** **(%)**	**OR** **(95% CI)**	***p* Value**	**n/N** **(%)**	**OR** **(95% CI)**	***p* Value**
**hrHPV DNA**						
**−**	10/21 (47.6)	Ref		1/21 (4.8)	Ref	
**+**	66/82 (80.5)	4.54 (1.64–12.53)	0.004	22/82 (26.8)	7.33 (0.93–57.9)	0.06
**HPV16/18 DNA**						
**−**	34/53 (64.2)	Ref		6/53 (11.3)	Ref	
**+**	42/50 (84.0)	2.93 (1.14–7.52)	0.025	17/50 (34.0)	4.03 (1.44–11.32)	0.008
**hrHPV mRNA**						
**−**	12/27 (44.4)	Ref		1/27 (3.7)	Ref	
**+**	64/76 (84.2)	6.67 (2.51–17.73)	0.0001	22/76 (28.9)	10.59 (1.35–82.94)	0.02
**p16/Ki-67**						
**−**	19/40 (47.5)	Ref		1/40 (2.5)	Ref	
**+**	57/63 (90.5)	10.50 (3.69–29.90)	<0.0001	22/63 (34.9)	20.93 (2.69–162.79)	0.004

ASCUS+: ASCUS (atypical squamous cells of undetermined significance) and LSIL (low-grade squamous intraepithelial lesion) and HSIL (high-grade squamous intraepithelial lesion); CI; Confidence Interval; OR, Odds Ratio; Ref, reference.

**Table 4 pathogens-10-00888-t004:** Association of ASCUS+ and HSIL cytology with the biomarker combinations in (A) HIV-uninfected and (B) HIV-infected MSM.

**(A)**	**HIV-Uninfected MSM**					
	**ASCUS+**			**HSIL**		
	**n/N** **(%)**	**OR** **(95% CI)**	***p* Value**	**n/N** **(%)**	**OR** **(95% CI)**	***p* Value**
**hrHPV DNA/hrHPV mRNA**						
−/−	6/10 (60.0)	Ref		2/10 (20.0)	Ref	
−/+; +/−	5/7 (71.4)	1.67 (0.21–13.22)	0.63	1/7 (14.3)	0.67 (0.05–9.19)	0.76
+/+	22/30 (73.3)	1.83 (0.41–8.23)	0.43	10/30 (33.3)	2.00 (0.36–11.23)	0.43
**hrHPV DNA/p16/Ki-67**						
−/−	5/8 (62.5)	Ref		1/8 (12.5)	Ref	
−/+; +/−	8/14 (57.1)	0.80 (0.13–4.74)	0.81	2/14 (14.3)	1.17 (0.09–15.32)	0.91
+/+	18/23 (78.3)	2.16 (0.38–12.32)	0.39	10/23 (43.5)	5.38 (0.57–51.17)	0.14
**HPV16/18 DNA/hrHPV mRNA**						
−/−	8/14 (57.1)	Ref		3/14 (21.4)	Ref	
−/+; +/−	12/14 (85.7)	4.50 (0.72–28.15)	0.11	4/14 (28.6)	1.47 (0.26–8.23)	0.66
+/+	13/19 (68.4)	1.63 (0.39–6.82)	0.51	6/19 (31.6)	1.69 (0.34–8.40)	0.52
**HPV16/18 DNA/p16/Ki-67**						
−/−	8/13 (61.5)	Ref		2/13 (15.4)	Ref	
−/+; +/−	13/19 (68.4)	1.35 (0.31–5.94)	0.69	5/19 (26.3)	1.96 (0.32–12.12)	0.47
+/+	10/13 (76.9)	2.08 (0.38–11.48)	0.40	6/13 (46.2)	4.71 (0.73–30.28)	0.10
**hrHPV mRNA/p16/Ki-67**						
−/−	7/11 (63.6)	Ref		2/11 (18.2)	Ref	
−/+; +/−	5/11 (45.5)	0.48 (0.09–2.63)	0.40	1/11 (9.1)	0.45 (0.03–5.84)	0.54
+/+	19/23 (82.6)	2.71 (0.53–13.92)	0.23	10/23 (43.5)	3.46 (0.61–19.72)	0.16
**(B)**	**HIV-Infected MSM**					
	**ASCUS+**			**HSIL**		
	**n/N** **(%)**	**OR** **(95% CI)**	***p* Value**	**n/N** **(%)**	**OR** **(95% CI)**	***p* Value**
**hrHPV DNA/hrHPV mRNA**						
−/−	9/20 (45.0)	Ref		1/20 (5.0)	Ref	
−/+; +/−	4/8 (50.0)	1.22 (0.24–6.31)	0.81	0/8 (0.0)	0.76 (0.03–20.74)	0.87
+/+	63/75 (84.0)	6.42 (2.19–18.81)	<0.001	22/75 (29.3)	7.89 (0.99–62.59)	0.05
**hrHPV DNA/p16/Ki-67**						
−/−	6/16 (37.5)	Ref		0/16 (0.0)	Ref	
−/+; +/−	17/29 (58.6)	2.36 (0.67–8.27)	0.18	2/29 (6.9)	3.00 (0.13–66.40)	0.49
+/+	53/58 (91.4)	17.77 (4.51–69.23)	<0.001	21/58 (36.2)	18.92 (1.08–331.39)	0.044
**HPV16/18 DNA/hrHPV mRNA**						
−/−	11/25 (44.0)	Ref		1/25 (4.0)	Ref	
−/+; +/−	24/30 (80.0)	5.09 (1.54–16.79)	0.008	5/30 (16.7)	4.80 (0.52–44.15)	0.17
+/+	41/48 (85.4)	7.45 (2.42–22.97)	<0.001	17/48 (35.4)	13.16 (1.63–106.00)	0.016
**HPV16/18 DNA/p16/Ki-67**						
−/−	13/28 (46.4)	Ref		0/28 (0.0)	Ref	
−/+; +/−	27/37 (73.0)	3.11 (1.10–8.80)	0.03	7/37 (18.9)	14.02 (0.77–256.77)	0.08
+/+	36/38 (94.7)	20.77 (4.17–103.49)	<0.001	16/22 (42.1)	41.80 (2.38–735.26)	0.011
**hrHPV mRNA/p16/Ki-67**						
−/−	8/22 (36.4)	Ref		0/22 (0.0)	Ref	
−/+; +/−	15/23 (65.2)	3.28 (0.97–11.13)	0.06	2/23 (8.7)	5.23 (0.24–115.39)	0.29
+/+	53/58 (91.4)	18.55 (5.25–65.60)	<0.0001	21/58 (36.2)	25.80 (1.49–447.01)	0.026

ASCUS+: ASCUS (atypical squamous cells of undetermined significance) and LSIL (low-grade squamous intraepithelial lesion) and HSIL (high-grade squamous intraepithelial lesion); CI; Confidence Interval; OR, Odds Ratio; Ref, reference.

## Data Availability

The data presented in this study are available on request from the corresponding author. The data are not publicly available due to the fact that they include personal data from vulnerable populations.

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
