# Peer review of "Evaluation of HPV-Related Biomarkers in Anal Cytological Samples from HIV-Uninfected and HIV-Infected MSM"

_pathogens, 2021, doi:10.3390/pathogens10070888_

Round 1
Reviewer 1 Report
In this manuscript the authors evaluated several HPV-based biomarkers in anal samples of HIV-infected and HIV-uninfected individuals.
Despite the lower number of HIV-uninfected patients is a limitation of the study, findings reported here are novel and interesting, since they point to p16/Ki67 as surrogate biomarker for the detection of anal lesions. The article is well written, tables are clear and self-explanatory, and the discussion is well balanced.
Author Response
In this manuscript the authors evaluated several HPV-based biomarkers in anal samples of HIV-infected and HIV-uninfected individuals.
Despite the lower number of HIV-uninfected patients is a limitation of the study, findings reported here are novel and interesting, since they point to p16/Ki67 as surrogate biomarker for the detection of anal lesions. The article is well written, tables are clear and self-explanatory, and the discussion is well balanced.
Authors’ response: we thank the reviewer for appreciating our work.
Reviewer 2 Report
I recommend that the authors simplify the abstract and just focus on the question of the study, the results and interpretation without using numbers and percentages, which do not mean anything without reading the entire manuscript. I also suggest that the abbreviations in the abstract without explanation should be avoided: what are NILM, ASCUS, LSIL? Also, it has never been mentioned why p16/Ki-67 marker was chosen for the screening.
Author Response
I recommend that the authors simplify the abstract and just focus on the question of the study, the results and interpretation without using numbers and percentages, which do not mean anything without reading the entire manuscript. I also suggest that the abbreviations in the abstract without explanation should be avoided: what are NILM, ASCUS, LSIL? Also, it has never been mentioned why p16/Ki-67 marker was chosen for the screening.
Authors’ response: we simplified the Abstract, also deleting the abbreviations we could not explain due to the word limit (line 24). Regarding p16/Ki-67, an explanation of the significance of this biomarker has now been added to the Introduction (page 2).
Reviewer 3 Report
Dear Authors,
this is an interesting investigation of HPV related biomarkers in relation to cytological abnormalities in MSM. The main drawbacks of the study are the small study population and the lack of multivariate analysis. The authors could try to apply multivariate methods to investigate possible patterns of biomarkers associated with the cytologic abnormalities.
In addition, the following points should be considered to improve the manuscript.
L20: The meaning of “HPV16/18” within the enumeration is not quite clear – is it HPV-type or type 16/18 DNA..?
Check for correct use of official gene/protein nomenclature - e.g. “p16” needs to be introduced as an abbreviation for “Cyclin-dependent kinase inhibitor 2A” or “p16-INK4a”
Check that all abbreviations were introduced at first mention – e.g. Abstract: NILM, ASCUS,.. (some abbreviations have been explained in the "MM" section, possibly due to a change in journal style, but these should be moved up in the manuscript)
L26: specify “both populations” (from reading the abstract it is only one population to here), also using “mRNA” here is confusing: mRNA transcribed from which gene? Is “mRNA” specific for HPV16 genotype?
Tables: The terms "hrHPV-DNA" in conjunction with "HPV16/18-DNA" are somewhat confusing and ambiguous. Does the former mean "all hrHPV DNA" or " hrHPV DNA other than HPV16/18"? Please check other tables as well; Without specifying the abbreviations, it remains unclear to the reader that the grade increases from NILM>ASCUS ..
Table 2: ki67 > Ki67; please provide the p-values in the table instead of the text; which statistical test was applied?
L107: The term “HPV16 infection” is not correct in this regard, since DNA-detection is compared to mRNA and (cellular) protein expression. Since HPV "infection" is a past event, and is not directly detected here, I would recommend not using this term but to be specific about the markers.
L115: specify “ASCUS+”, also in Table 3.
Table 3 is quite full and confusing - is it possible to display this data differently, possibly graphically. If the authors decide to keep a table, separate columns should be included for the p-values (also the non-significant ones) (currently I can't track what was compared to which test), and perhaps take the total number out of each of the cells and put it elsewhere.
The classification positive/negative of the biomarker combination is not clear. Especially when using the term "and/or" for the combination. In any case, there can be four combinations: -/-, -/+. +/-, and +/+ and it is not clear from the table what positive/negative means.
Section 4.5 contains information about HPV classification which I would recommend to move to the HPV-testing section (4.2).
MM: No information on HPV 16/18 typing is provided - as indicated, the Aptima HPV assay only provides information on 14 HR types "as a pooled result".
L137: “Conversely” is not correct since there is no converse effect.
L164-166: I don't think you can say from the data “that hrHPV mRNA and p16/Ki-67 detection is not associated to HPV16 infection” – There is an association of non-16-HR-HPV DNA detection and mRNA and p16/Ki-67 detection as it is for HPV16. However, the strength of this association appears to be not stronger/different for HPV16 than for non-16-HR-HPV types.
Author Response
Dear Authors,
this is an interesting investigation of HPV related biomarkers in relation to cytological abnormalities in MSM. The main drawbacks of the study are the small study population and the lack of multivariate analysis. The authors could try to apply multivariate methods to investigate possible patterns of biomarkers associated with the cytologic abnormalities.
Authors’ response: We thank the reviewer for the precise comments, which have allowed us to make the manuscript clearer to the readers.
Regarding the analysis requested, we apologize but it is not completely clear to us. Our study was designed for evaluating the associations between HPV-related biomarkers and cytologic abnormalities. To this aim, we also investigated possible patterns combining in pairs the biomarkers under study (as previously shown in Table 3). To dissect further this aspect, we have now included a new table (Table 4) that shows, for each pair of biomarkers, the degree of association with cytologic abnormalities for the cases positive for any of the biomarkers of the couple and, separately, for those showing simultaneous positivity. These new data have been detailed in the Results (page 7) and discussed (page 11). M&M have been modified accordingly (page 13). We hope that this new version meets your request.
In addition, the following points should be considered to improve the manuscript.
L20: The meaning of “HPV16/18” within the enumeration is not quite clear – is it HPV-type or type 16/18 DNA..?
Authors’ response: “DNA” has been added to specify that we referred to HPV16/18 DNA (Abstract, line 19). This was already specified throughout the text.
Check for correct use of official gene/protein nomenclature - e.g. “p16” needs to be introduced as an abbreviation for “Cyclin-dependent kinase inhibitor 2A” or “p16-INK4a”
Authors’ response: we have now specified the correct name for p16 (Introduction, page 2).
Check that all abbreviations were introduced at first mention – e.g. Abstract: NILM, ASCUS,.. (some abbreviations have been explained in the "MM" section, possibly due to a change in journal style, but these should be moved up in the manuscript)
Authors’ response: We have checked all the abbreviations and introduced them at first mention. Regarding the Abstract, we have now rephrased the sentence, specifying the meaning of the abbreviations for the cytological categories mentioned in this section.
L26: specify “both populations” (from reading the abstract it is only one population to here), also using “mRNA” here is confusing: mRNA transcribed from which gene? Is “mRNA” specific for HPV16 genotype?
Authors’ response: We have now specified that our study included HIV-uninfected and infected MSM (Abstract, lines 20-21). We have also specified that mRNA referred to hrHPV E6/E7 mRNA (Abstract, lines 20, 23 and 27).
Tables: The terms "hrHPV-DNA" in conjunction with "HPV16/18-DNA" are somewhat confusing and ambiguous. Does the former mean "all hrHPV DNA" or " hrHPV DNA other than HPV16/18"? Please check other tables as well; Without specifying the abbreviations, it remains unclear to the reader that the grade increases from NILM>ASCUS.
Authors’ response: "hrHPV-DNA" means "all hrHPV DNA", as specified in M&M (page 12, where all the types included in this group have been listed). Regarding the cytological categories, the abbreviations have been now introduced at first mention (Introduction, page 2) and have been also specified in the Table footnotes.
Table 2: ki67 > Ki67; please provide the p-values in the table instead of the text; which statistical test was applied?
Authors’ response: “ki67” has been replaced by Ki-67 as in the rest of the manuscript; p-values have been provided in the Table and deleted from the text; the test of significance has been now specified in the M&M (page 13).
L107: The term “HPV16 infection” is not correct in this regard, since DNA-detection is compared to mRNA and (cellular) protein expression. Since HPV "infection" is a past event, and is not directly detected here, I would recommend not using this term but to be specific about the markers.
Authors’ response: we agree with this comment, and also based on the comment on line 164-166, we rephrased the sentence as follows “Presence of HPV16 DNA did not display a significantly different association with hrHPV mRNA or p16/Ki-67 positivity compared to the presence of DNA of hrHPVs other than HPV16”.
L115: specify “ASCUS+”, also in Table 3.
Authors’ response: this has been specified.
Table 3 is quite full and confusing - is it possible to display this data differently, possibly graphically. If the authors decide to keep a table, separate columns should be included for the p-values (also the non-significant ones) (currently I can't track what was compared to which test), and perhaps take the total number out of each of the cells and put it elsewhere.
Authors’ response: in order to make this Table more readable, we split it into Table 3A (for HIV-uninfected MSM) and Table 3B (for HIV-infected MSM); we preferred to keep the absolute numbers for completeness; we have now added the p values for each association and moved the part regarding the combinations to Table 4.
The classification positive/negative of the biomarker combination is not clear. Especially when using the term "and/or" for the combination. In any case, there can be four combinations: -/-, -/+. +/-, and +/+ and it is not clear from the table what positive/negative means.
Authors’ response: we have now displayed the exact classification of the results for each combination.
Section 4.5 contains information about HPV classification which I would recommend to move to the HPV-testing section (4.2).
Authors’ response: we have moved the information to section 4.2 as suggested.
MM: No information on HPV 16/18 typing is provided - as indicated, the Aptima HPV assay only provides information on 14 HR types "as a pooled result".
Authors’ response: information regarding HPV 16/18 only concerns DNA, not mRNA; in all the Tables as well as throughout the text, it has been specified that HPV 16/18 always refers to HPV 16/18 DNA.
L137: “Conversely” is not correct since there is no converse effect.
Authors’ response: we have replaced “Conversely” with “Differently”.
L164-166: I don't think you can say from the data “that hrHPV mRNA and p16/Ki-67 detection is not associated to HPV16 infection” – There is an association of non-16-HR-HPV DNA detection and mRNA and p16/Ki-67 detection as it is for HPV16. However, the strength of this association appears to be not stronger/different for HPV16 than for non-16-HR-HPV types.
Authors’ response: thank you for this comment; we agree with it, thus we have now rephrased this sentence as follows “The strength of the association of HPV16 with hrHPV mRNA and p16/Ki-67 positivity did not appear to be significantly different compared to hrHPV types other than HPV16”.
Round 2
Reviewer 3 Report
No further comments or questions.